# DMA Analysis of Plasma Modified PVC Films and the Nature of Initiated Surface Changes

**DOI:** 10.3390/ma15134658

**Published:** 2022-07-02

**Authors:** Róbert Janík, Marcel Kohutiar, Andrej Dubec, Maroš Eckert, Katarína Moricová, Mariana Pajtášová, Darina Ondrušová, Michal Krbata

**Affiliations:** 1Faculty of Industrial Technologies in Púchov, Alexander Dubček University of Trenčín, Ivana Krasku 491/30, 020 01 Púchov, Slovakia; andrej.dubec@tnuni.sk (A.D.); katarina.moricova@tnuni.sk (K.M.); mariana.pajtasova@tnuni.sk (M.P.); darina.ondrusova@tnuni.sk (D.O.); 2Faculty of Special Technology, Alexander Dubcek University of Trenčín, Ku Kyselke 469, 911 06 Trenčín, Slovakia; marcel.kohutiar@tnuni.sk (M.K.); maros.eckert@tnuni.sk (M.E.); michal.krbata@tnuni.sk (M.K.)

**Keywords:** dynamic mechanical analysis, polymer foil, glass transition temperatures, DCSBD plasma discharge

## Abstract

The application of DCSBD (Diffuse Coplanar Surface Barrier Discharge) plasma is referred to as the surface modification/activation of materials. The exposure of material surfaces to DCSBD plasma is initiated by changes in their chemical composition, surface wettability and roughness. The given study presents the mentioned plasma application in the context of the modification of the material viscoelastic properties, namely the PVC polymer film. The measurement of viscoelastic properties changes of PVC was primarily examined by a sensitive thermal method of dynamic-mechanical analysis. This analysis allows identifying changes in the glass transition temperature of PVC, before and after DCSBD plasma application, Tangens Delta, supported by glass transition temperatures of Elastic and Loss modulus. The results of the present study prove that DCSBD plasma applied on both sides to PVC surfaces causes changes in its viscoelastic properties. In addition, these changes are presented depending on the variability of the material position, with respect to the winding of the electrodes in the ceramic dielectric generating the DCSBD plasma during modification. The variability of the PVC position holds an important role, as it determines the proportion of filamentous and diffuse components of the plasma that will interact with the material surface during modification. The application of DCSBD plasma must, therefore, be considered a complex modification of the material, and as a result, non-surface changes must also be considered.

## 1. Introduction

In the present work, the effect of PVC polymer film modification by DCSBD plasma (Diffuse Coplanar Surface Barrier Discharge) is investigated. The plasma is generated on the surface by the ceramic dielectric in the form of H-shaped micro-discharges, where there is a distinct filament in the space between two adjacent band electrodes and two areas of diffuse plasma above the electrodes [1,2,3,4,5,6,7,8]. As the input power increases, the density of micro discharges increases and the proportion of diffuse plasma increases compared to the filamentary plasma, which guarantees greater homogeneity of the surface treatment of flat or flexible materials. DCSBD plasma provides a high-power density of the generated plasma since it burns at the dielectric surface in a 0.3 mm thin layer [9]. The aim of the work is to observe the facts that are directly initiated by such a modification, preferably at the level of surface changes. The changes that cause the surface modification (in some cases also referred to as activation) are interpreted in the present work based on the results of the contact angle changes [8], scanning electron microscopy analysis (SEM) [8,10], atomic force microscopy analysis (AFM) [10] and X-ray photoelectron spectroscopy analysis (XPS) [10,11,12]. The obtained results are presented in the context of thermal analyses, dynamic mechanical analysis (DMA) and differential scanning calorimetry (DSC). In the thermal analysis, changes in the glass transition temperature are attributed to the glass transition temperatures before and after exposure of the polymer films to the DCSBD plasma.

In recent years, research has been conducted into the use of plasma in modifying a wide range of materials; plastics and glass, composites, fibers, textiles, films and membranes, wood, food (packaging and sterilization) and other materials [10,11]. Murthy et al. proved that open air plasma treatment increases the bond strength of the glass-PP, neat PP and HDPE surfaces, by increasing their surface energies [13]. Amorim et al. studied the effects of cold plasma treatment on amorphous hydrogenated carbon (FA), polypropylene (PP) and polystyrene (PS) [14]. Bagiatis et al. studied the effect of atmospheric pressure plasma treatment (APPT) on polymethyl methacrylate (PMMA) substrates, which were investigated in terms of both the chemical and topographical changes introduced to the polymer surface and its influence on PMMA-to-glass adhesion [15]. Buček et al. examined the effect of DCSBD air plasma treatment on the adhesive strength of glass bonds. The plasma was found to improve the overall strength of the bonds, resulting in an increase in the adhesive bond strength [16]. Homola et al. showed that the DCSBD plasma treatment had a significant effect on the wettability of glass surfaces [17]. Rafailovič et al. proved that carbon fiber reinforced composites (CFRP) can be activated by an atmospheric pressure plasma source to improve the performance of the galvanically plated Cu layer on its surface [18]. Shepa et al. studied the impact of plasma treatment on the different atmospheres on the surface of the polymer TiO2/PVP microfibers, prepared by needleless electrospinning technology [19]. Pavliňák et al. studied nanofiber carriers, polypropylene nonwoven textile and paper, which were pretreated by dielectric barrier discharge in continuous mode to improve the adhesion between the produced nanofibers and substrate [20]. Trejbal et al. demonstrated the influence of plasma treatment on the surface properties of PET fibers used as micro reinforcement in cementitious composites [21]. Radić et al. showed the plasma activation of nonwoven polypropylene (PP) using two different ambient air plasma sources: volume dielectric barrier discharge (DBD) and diffuse coplanar surface barrier discharge (DCSBD) and its functionalization by silver ion deposition [22]. Naebe et al. summarized the rationale of plasma use in textile antimicrobial finishing through a critical analysis of recent studies and emphasizes the types and mechanisms of plasma techniques available for application [23]. Hasan et al. described melt blown-nonwoven polypropylene (NW-PP) membranes, which were functionalized using (O2) cold plasma for the conversion of highly hydrophobic properties into hydrophilic properties to enhance the absorption and adhesion of graphene films on the fiber surface [24]. Kawakami et al. treated polypropylene (PP) film surfaces using air-based nonequilibrium atmospheric pressure plasma jets generated with a twisted wires-cylindrical electrode configuration and compared them with PP samples treated with Ar plasma jets [25]. Correia et al. report on the modification of the surface wettability of poly (vinylidene fluoride) (PVDF) and PVDF copolymer films and membranes by plasma treatments under different conditions, i.e., oxygen and argon atmospheres [26]. Talviste et al. characterized the surface changes of several species of thermally modified wood after plasma treatment with diffuse coplanar surface barrier discharge (DCSBD) [27]. Macedo et al. proved plasma treatment as a feasible approach for surface activation of kapok fibers, thus improving matrix/filler adhesion [28]. Bulbul et al. studied the effect of cold plasma on the properties of xanthan gum (XG) at different power and treatment times [29]. Limnaios et al. evaluated diffuse coplanar surface barrier discharge cold atmospheric pressure plasma (DCSBD CAPP) treatment as a potential decontamination technique for red currants and investigated its effects on microbial load and quality [30]. Hu et al. demonstrated that atmospheric cold plasma treatment is a promising postharvest strategy to control the natural decay and meanwhile maintain the fruit quality during 10-d storage at room temperature [31]. Hou et al. presented the effects of cold plasma technology on the quality of blueberry juice [32]. Misra et al. presented a review of the design aspects of embedded plasma systems, and packaging requirements and discussed their effectiveness with respect to microbiological and chemical food safety [33]. Krumpolec et al. presented a method for fast cleaning and activation of silicon wafers at atmospheric pressure by a novel low-temperature plasma generator called multi-hollow surface dielectric barrier discharge MSDBD [34]. Hvojnik et al. investigated and compared rapid ambient air atmospheric plasma treatment with a time-consuming standard cleaning procedure for FTO substrates [35]. Medvecká et al. conducted plasma-assisted calcination using a low-temperature plasma generated by Diffuse Coplanar Surface Barrier Discharge in ambient air and it was studied for the preparation of alumina nanofibers from polyvinyl pyrrolidone/aluminum butoxide fibers prepared by electrospinning [36].

The main idea of the present study is to investigate changes on PVC surfaces after DCSBD plasma treatment. This change/modification can be initiated by a generated DCSBD plasma discharge directly on the investigated surfaces or an indirect recorded change in the averted surface (backside treatment), possibly supported by the effects of the generated DCSBD discharge, i.e., UV radiation, ozone generated [37], and electric discharge generated. After plasma treatment, new, matte areas on the surfaces of PVC polymeric films appear. Can these areas contribute to the mechanical damage of thin PVC polymer films? This idea has been expressed in scientific work [38]. The method of DCSBD plasma discharge modification was, therefore, designed in the experimental part to assume such a change, which could be attributed to changes in viscoelastic properties of PVC film, which can be quantified based on the changes in glass transition temperature (PVC exposed to DCSBD plasma for an extended time in static mode). Using this setup, four basic experimental series of PVC polymer films were prepared in a plasma reactor. One sample, modified by DCSBD discharge, was also subjected to a standard static tensile test to determine its strength and elongation. In a research article [12], the authors argue that DCSBD plasma improves the adhesion of polymer surfaces (e.g., Virgin polyamide 6-PA 6), while exposure times longer than 30 s are no longer recommended for industrial use. The present work evaluates the longer exposure time of the polymer film PVC (60 s) DCSBD plasma for possible industrial applications. The work presents findings at the level of initiated surface changes when no free space was left between the investigated material and the surface of the ceramic dielectric and quantifies possible changes in internal property changes in the glass transition temperature of the PVC polymer film. At the level of surface changes, the work provides evidence of surface modifications that should not have been modified because they were not oriented to a homogeneous plasma layer generated on a ceramic dielectric during the process. The ceramic dielectric contains metal electrodes, just above which the diffuse part of the electric discharge is formed and the main location of the diffuse plasma.

## 2. Materials and Methods

Five series of experimental samples of PVC polymer film were prepared for the experiment on a pneumatic die cutting machine in a standardized size and shape referred to as type 1B. Type 1B is the shape of a test body (in this case the shape of blades), the dimensions of which are prescribed by the standards EN ISO 527-3. Selected test samples for four series of PVC film were modified using DCSBD plasma (signatures: I, II, III and IV). The investigated polymer is a PVC foil standardly used in office applications. The sample signature (PVC) was not surface modified using DCSBD plasma treatment. Each of those series (PVC, non-plasma treated and I, II, III, IV, plasma treated) contained 10 samples. The setting of the experimental series corresponds to the possibilities of the sample guide position through the DCSBD plasma layer.

The thickness of the polymer film was 0.11 ± 0.02 mm. Selected test samples for four series of PVC film were surface modified using DCSBD plasma. The investigated polymer is a transparent, glossy PVC foil (PRESTIGE 200 mic, Univox, CZ). The maximum application temperature range of rigid PVC is in the temperature range of 50–80 °C, while the glass transition temperature is in the temperature range of 60–100 °C [39]. 

DCSBD plasma was generated on a ceramic dielectric (manufactured by Roplass s.r.o., Brno, CR) of plasma reactor KPR 200 (Chemosvit a. s, Svit, SR) mm at 350 W with an exposure time of 60 s = surface of polymeric film PVC (Figure 1). 

The modification time may seem unusually long, but according to the available literature, it is suitably set to be able to observe new phenomena of changes/expected degradation [10]. The safety of the process (temperature of the ceramic dielectric and PVC polymer films) was monitored using a thermal imaging camera Testo 868 (TESTO, MY). The experimental method DCSBD is described in the literature [40]. As a result of the DCSBD plasma discharge effect on the polymer film surfaces, new matte areas were visible in contrast to the originally glossy and smooth PVC polymer films. Since the KPR 20 plasma reactor device does not contain a movable element for holding samples as described in other scientific studies [41,42,43,44,45], it was possible, in combination with the long exposure, to discharge to create characteristic white areas on the PVC polymer film surface. The stated position of the micro-discharges and their movement through the ceramic dielectric according to electrode windings can also be found in the literature [38] and in Figure 2. 

The subject of the investigation was a sample of polymeric PVC film without plasma treatment and polymeric PVC films exposed to direct plasma treatment according to exposure against electrode windings in a ceramic dielectric, designated as I, II, III and IV (Figure 3). The position of the PVC on the ceramic dielectric and the PVC modification procedure is shown in Figure 1.

To determine the contact angle changes between the test liquid and the surface of polymeric films (without DCSBD plasma modification: PVC and exposed to DCSBD plasma modification: I, II, III, IV) the Fowkes method was used for the experiment. Two liquids (polar and non-polar: dispersive) are used in this method. Distilled water (DW) was used as the polar liquid and diiodomethane (MI) was used as the non-polar liquid. Surface energy values were calculated from the measured values of the contact angles between the liquid and the samples of polymer films. The contact angles between the liquids and the solid surface of the standard PVC sample and the plasma modified films I–IV were determined using the sessile drop method. Using a micropipette, 10 drops of the appropriate liquid with a volume of 10 μL were placed on the surface of the PVC samples. A CCD camera was focused on the profile of each drop. Using uEye software, drop profiles were scanned, and contact angles were evaluated in ImageJ software. The profile of the drops was manually marked on the scanned profiles in the ImageJ software. After marking the profile of the drop, the contact angle of the drop on the PVC sample surface was evaluated. Distilled water (DW) with a predominantly polar component (γ_l_^d^ = 21.8 mJ·m^−^^2^, γ_l_^p^ = 51.0 mJ·m^−^^2^) and diiodomethane (MI) as a dispersing component (γ_l_ = γ_l_^d^ = 50.8 mJ·m^−^^2^) were used as test liquids [46,47].

Topography measurements of the studied PVC samples, their surface roughness and surface area were determined using an atomic force microscope MFP-3D Infinity (Oxford Instruments, Abingdon-on-Thames, UK). The measurements were performed in a mode where the tip is not in constant contact with the sample surface, thus reducing the risk of damage to the tip or the measured sample surface (AC Tapping mode in Air). The tip designation is AC 160TS. The sizes of the observed sample areas were 20 × 20 µm. To obtain a more detailed topography of the PVC polymer films, some areas were scanned at a higher resolution (5 × 5 μm). Each sample was measured five times, and the average value of the surface area increase, and area roughness was recorded. Observing the opposite side of the sample, this work is the first to record and describe backside treatment. Therefore, the resolution of the images from a larger area was used for better display.

The changes induced by DCSBD plasma treatment on the surface of the studied PVC samples were also monitored using a Tescan Vega3 thermoemission scanning electron microscope (Tescan Company, Brno, CZ) in the secondary electron (SE) mode with the applied accelerating voltage (HV) set at 20 to 30 kV. Prior to the observation, the samples were pre-treated with an Ecoson ultrasonic cleaner in isopropyl alcohol, then dried with a stream of nitrogen and attached to a metal support with carbon tape. When observing the unmodified PVC surface, a magnification of 2000× and 15,000× was sufficient. In addition, 6000× and 50,000× magnifications were used to observe plasma modified surfaces. 

X-ray photoelectron spectroscopy analysis was performed using the thermo scientific system K-Alpha XPS (Thermo Fisher Scientific, Oxford, UK), on the premises of the Slovak Academy of Sciences in Bratislava. The spectrometer is equipped with a micro-focused monochromatic X-ray source Al Kα (1486.6 eV). A 400 µm X-ray beam was used at 6 mA × 12 kV. The spectra for examination were obtained in the constant energy mode of the analyzer with transmission energy of 200 eV. Narrow regions were collected using an image acquisition mode (150 eV transition energy), which enabled fast data collection (5 s per region). Thermo Scientific Avantage software, version 5.9922 (Thermo Fisher Scientific) was used for digital data acquisition and processing. Spectral calibration was determined using an automated calibration routine and internal Au, Ag, and Cu standards supplied with the K-Alpha system. Surface composition (atomic%) was determined by considering the integration areas of the detected atoms peaks and the respective sensitivity factors.

Thermal analysis was performed on a TA Instruments Q800 dynamic-mechanical analysis device (DMA Q800, TA Instruments, New Castle, DE, USA). The test was performed by the tensile film test method at a frequency of 1 and 10 Hz. For each series of polymers, 10 samples were prepared and analyzed, for a total of 50 samples. The test temperature range was set from −25 to 125 °C. Software Universal Analysis 2000 for Windows 2000/XP/Vista; Version 4.5A Build 4.5.0.5 was used to evaluate the results of the DMA analysis. The following properties were evaluated: Storage modulus (E′), Loss modulus (E″) and Tangens Delta (Tan δ). The glass transition temperatures were determined from 10 measured curves for both modules and the Loss factor for all five series of PVC polymer film. Together, 150 DMA measurement records were evaluated for the purpose of determining the glass transition temperatures. The average values were processed into tables, while the resulting values are based only on the average of a group of numbers that did not contain values marked as outliers.

DSC analysis was performed using a Mettler Toledo TGA/DSC 2 (Mettler-Toledo, Columbus, OH, USA) instrument, in an oxygen atmosphere with a flow rate of 20 mL.min^−^^1^, a temperature interval from 30 to 90 °C, and a heating rate of 3 °C.min^−^^1^. The results were processed using STARe Thermal Analysis software. “Midpoint” values were measured for the glass transition values. The DSC instrument was calibrated based on the onset melting temperature and heat of Indium fusion according to the literature [48].

The static tensile test of PVC samples was performed on a universal Autograph AG-X plus tearing machine from Shimadzu (Shimadzu Corporation, Kyoto, Japan), according to ISO 527-3: 2018 [49]. The PVC samples were cut into the dumbbell-shaped geometry type 1B. The results were processed using Trapezium X software v. 1.4.5. The samples were attached by tearing machine jaws the set speed of their movement was 200 mm.min^−^^1^. During the test, the sample was gradually stretched until the acting force exceeded the strength limit of the material when the sample subsequently broke.

## 3. Results and Discussions

### 3.1. Study of Surface Properties

#### 3.1.1. Contact Angles Observation 

The measured values of the polymer PVC film surface contact angles with the test liquids diiodomethane and distilled water, according to Fowkes, are shown in Table 1.

##### Comparison of Contact Angle Changes in PVC, I, II, III, IV Samples

By evaluating the value of the standard deviations, all exposures of the investigated types of plasma modification and the homogeneity of the plasma modification can be determined. The homogeneity of the PVC film surface modification is best in most of the tested types in the type III modification when in most measurements the standard deviations reached the lowest values. It can be stated that with plasma exposure type I and II, the surface properties of the PVC polymer film change more rapidly.

It can be stated that with all types of plasma exposure (I, II, III and IV) and treated polymer films, the surface of the PVC polymer films became more hydrophilic after modification. The type IV sample could not be prepared for comparison in MONO and INDIRECT MONO, because the cross pattern is formed only when the sample is the rotated DUAL type. This pattern probably best distributed the initiated surface changes and can be thought to have promoted the formation of a new, more homogeneous surface layer. The decrease in contact angle (up to 13.5°) in the MI test fluid was most pronounced. This decrease could be due to some synergy after the material was turned on the ceramic dielectric, or by a hitherto undescribed action. 

There is a presumption that the degradation of plastics can be realized mainly through radical mechanisms initiated by singlet oxygen or a hydroxyl radical. With the assistance of oxygen, PVC dehydrochlorination takes place, a reaction in which hydrogen and chlorine are split off from the structure of the original polymer and a double bond is formed between the respective carbons [10,39].

##### Comparison of Contact Angle Changes in Type1/Type 2 Modifications

By comparing the changes in the values of the contact angles using the DUAL/MONO/INDIRECT MONO modification method, it can be stated that the DCSBD plasma modification of polymer films significantly changes the surface of this material. This finding is in accordance with the literature [50,51,52]. The contact angles of the plasma modified surfaces identically decreased significantly when using MI as a test liquid: from 36.5° (PVC) to an average value of 17.5° (DUAL), respectively, 30.5° (MONO) and 31.7° (INDIRECT MONO). A less significant decrease can be observed using the DW test liquid: from 88° (PVC) to 52.8° (DUAL), 49.8° (MONO) and 62.2° (INDIRECT MONO), respectively. Literature values for the water contact angle of PVC are 82 ± 2°. After surface modification, the contact angles decreased significantly, consistent with the presence of a hydrophilic material on the surface [50]. From the above results, it can be stated that the surface energy of PVC consists of a larger proportion of dispersive than polar components. This assumption was also confirmed by the calculated values of surface energy: PVC dispersive component = 35 mJ.m^−2^, polar component = 3.5 mJ.m^−2^, PVC+DCSBD dispersive component = 42.7 mJ.m^−2^ and polar component = 11.7 mJ.m^−2^. The observed changes in the proportions of the total surface free energy of the dispersive and polar components (due to induced oxygen-containing molecular groups) are thus in accordance with the literature [53,54]. 

Using the MONO modification type, the MI contact angle value reaches 28.9°. The largest change in the contact angle value (using MI as a test liquid) was observed at exposure IV = 13.5°. This value of the contact angle is almost a half lower. This finding is surprising and new because, in the DUAL modification type, the PVC surface was exposed to the same plasma treatment. The only variable that has changed in relation to the PVC surface is the exposure time over the ceramic dielectric generating the DCSBD plasma. This is because the DUAL modification type is implemented separately for each side of the PVC film. Figure 2 shows the DCSBD treatment types: Type 1 Direct exposures (MONO and DUAL).

Therefore, another plasma modification experiment was realized, where changes in the contact angles of the test liquids were observed on a surface that was never in direct contact with the generated DCSBD plasma. (INDIRECT MONO), Figure 2. The results obtained are again surprising and new: the surface of the PVC polymer film, which has never been in direct contact with the generated DCSBD plasma, showed changes in contact angles almost identical to those of direct surface modification (MONO). In the case of using the MI test liquid, the difference in the contact angle value is only 3.8%, while when using the DW test liquid, the difference is 24.9%. The values are shown in Table 1. The change in droplet profile when using each type of DUAL/MONO/Type 2 modification is shown in Figure 4.

There are basically three ways in which DCSBD plasma could modify a relatively thin polymeric material on the opposite surface, surface B:DCSBD plasma burns on the entire ceramic dielectric, while atoms, ions and free radicals are excited into the atmosphere from the free, unoccupied part of the ceramic dielectric. These can be released in the vicinity of the ceramic dielectric and subsequently fall on the opposite surface of the investigated PVC polymer film. Thus, a kind of plasma cloud with a kinetic potential may form above the surface of the ceramic dielectric (Figure 5).Insulated micro-discharge breakthroughs contribute to surface modification (Figure 6).DCSBD plasma, as it is generated, penetrates through the structure of the (thin polymer) material and thus affects the opposite surface (possible with the support and in combination with the generated UV, ozone [55] and heat from the ceramic dielectric).The fourth option may be a combination of the two previous hypotheses.

#### 3.1.2. Observation of Surfaces Using AFM Method 

The obtained results of the increase in the surface area and roughness for the unmodified/plasma modified surfaces of PVC are shown in Table 2. A standard PVC sample is characterized by the fact that it has a relatively uniform surface. Scratches and impurities were observed sporadically on it. The surface topography of the standard PVC sample can be seen in Figure 7.

The obtained results of the increased surface area and roughness for polymer films PVC modified by DCSBD plasma discharge realized for both sides (DUAL) in exposure types I, II, III and IV are shown in Table 2 and Figure 8.

##### Comparison of the Increase in Area and Roughness in PVC, I, II, III, IV Samples

Comparing the values of the increase in surface area and surface roughness, the average surface area increase of the polymer films after plasma exposure to DCSBD was 5.84% and the average increase in roughness was 28.94 nm. On average, all types of plasma DCSBD exposures are increases of 5.8% and 22.96 nm, respectively. However, exposure type III and in particular exposure type IV recorded the most significant and above-average changes in surface area increase as well as a change in roughness.

##### Comparison of Surface Area and Roughness Increase with Type1/Type2 Modifications

By observing the values of the increase in surface area and roughness on the surface A (type MONO) in Table 2, it can be stated that the reverse side of the PVC polymer film was modified in all types of exposure I, II, III and IV. AFM images of surfaces are shown in Figure 9. On average, after DCSBD plasma exposure, the increase in surface area was 4.17% and the roughness increased by 31.38 nm.

By comparing the values of the increase in surface area and roughness on surface B, it can be stated that the opposite side of the PVC polymer film was also modified (Table 2). Examination of side B for polymer films showed increases in area (especially for exposure types I and II), as well as an increase in roughness (again, for exposure types I and II). This is a new finding and modification can thus be described as indirect and may be undesirable in some applications. This modification may also be aided by the abnormally long exposure time of the DCSBD plasma on the polymer film. Therefore, we assume that at standard time exposures, the effect may not be so obvious. For yet unexplained reasons, exposure type III recorded below-average values for both monitored parameters. The values given are similar/close to the increase in area and even lower with the change in roughness than for the reference PVC sample, which was not modified by plasma. AFM images of surfaces are shown in Figure 10.

From the results obtained using AFM analysis, it can be stated that the surfaces of all the polymer films modified by DCSBD plasma discharge, recorded an increase in surface area and an increase in roughness (relative to the surface of the polymer film, which was not plasma modified) [56,57,58,59,60,61,62]. A new finding in the field of research is that a surface roughening of the plasma-unmodified side (surface B) on the PVC polymer film was noted, as well as an increase in surface area compared to the standard PVC sample. Based on the results obtained from the AFM method, another new finding is evident; the PVC polymer film was modified by plasma DCSBD, even though its distance was not always at an ideal distance from the ceramic dielectric, about 0.33 mm (KPR 20 plasma reactor equipment does not contain a movable element for holding the samples at a set distance from the ceramic dielectric generating the DCSBD plasma layer). Based on the above finding, it can be considered that the exposure time for samples treated with DCSBD plasma discharge from both sides (DUAL) can also be understood as the direct exposure time of surface A to the DCSBD plasma for 60 s (according to the scheme referred to as MONO) + the same indirect exposure time B, at that time of the opposite side 60 s (Figure 1). Subsequently, the sample was inverted so that the other side of the polymer film could be modified to complete the modification by DUAL exposure. As a result, the surfaces of the polymer films prepared by the DUAL modification type can be seen as the total exposure of the surfaces to the plasma above the ceramic dielectric: direct for surface A + indirect for surface B. This is followed by a rotation of the sample and direct exposure of the indirectly modified surface B to the direct exposure of the plasma + exposure of previously directly exposed surface A to the indirect exposure of the plasma. The resulting time thus doubles the exposure time of the whole sample above the electrode. A one-minute single exposure of the PVC polymer film to the surface of the DCSBD plasma-generating ceramic dielectric thus becomes a 2-min exposure (direct + indirect = DUAL).

#### 3.1.3. Elemental Changes on Surfaces Detected by XPS Analysis

The proportion of elements on the surfaces of PVC polymer films before and after exposure to plasma discharge (exposure types I, II, III and IV) was determined using X-ray photoelectron spectroscopy (XPS) analysis. The basic elements of the PVC polymer film are C, O and Cl. In addition, the existence of elements Na 1s = 0.8%, Ca 2p = 0.7%, N 1s = 0.8%, Si 2p = 0.4% and S 2p = 0.1% was measured. These can be marked as a type of contamination (e.g., by sample handling), while they are considered irrelevant in relation to the objectives of the given work. The DCSBD plasma formed characteristic, periodically repeating grid-like regions on the surface of the polymer, in which (originally still) clear and (new) matte white regions can be detected, in combination with the observed region. The change in the representation of the elements is, therefore, interesting not only according to the type of deposition on the plasma (I, II, III and IV) but also on the matte and clear areas formed on the surface of the polymer by the DCSBD plasma (Table 3).

##### Comparison of Change in PVC, I, II, III, IV Samples

Comparing the reference sample of PVC with other plasma modified samples in the observed area, which appears clear, then on the surface of the material after plasma treatment, the carbon decreased from 77.6% on average to 61.2%. This decrease may indicate the original contamination of the reference material surface that was removed by the DCSBD plasma discharge. When measuring the percentage of oxygen, an increase was recorded from 12.6% to an average of 18.1%. This finding is expected and is related to the fact that air was used as the working gas for the experiment. The incorporation of O atoms is related to the formation of functional polar groups on the surface of the material. Another situation occurred with chlorine, in which an increase from 7.1% to 18.5% can be observed. This may indicate that the DCSBD plasma modification has broken the C–Cl bonds, which may be related to the migration of chlorine to the surface of the investigated material.

The difference in the representation of the element’s occurrence on the surface of the plasma modified material can also be observed and evaluated in relation to the observed area with respect to the newly formed matte stripes on the polymer. These stripes are characterized by a lower percentage of Cl and C, and conversely, a higher percentage of O compared to clear areas. Thus, it was assumed that the intensity of the C–Cl bond decay and the formation of polar functional bonds on the matte surface proceeded at a higher rate than on the clear surface [63,64]. From this point of view, it is important in which position the sample is placed on the burning plasma, because the modification achieves two types of surfaces, each with a different material characteristic. This means that DCSBD plasma has a dual effect on the polymeric material. In terms of comparing the matte and clear areas between the types of polymer deposition on the burning plasma, types I, II, III and IV, differences between the combination of their material composition were noted. For type IV, this composition varied with lower intensity, which can be attributed to the specifics of this deposit. 

##### Comparison of Change in Type1/Type 2 Modifications

In the case of MONO type plasma modification with exposure type I, the elemental representation on the reverse side (surface B) of the polymer film was examined. The analysis values are shown in Table 3. The representation of C, O and Cl very conspicuously copies the average values from the DUAL type DCSBD plasma modification. The measured differences in the elemental composition are mostly at the level of tenths of a percent. The above finding provides direct evidence of the opposite surface (surface –B) modification of PVC polymer film, whose indirect modification by DCSBD plasma is presented in the work using the results obtained by AFM analysis and measured changes in contact angles.

#### 3.1.4. Surface Observation Using SEM Method

On the SEM images of the reference-plasma unmodified sample (Figure 11) of the PVC polymer film, the observed structure of the material appears to be homogeneous. On the surfaces, it is possible to observe a monotonous morphology with a rare occurrence of scratches or impurities.

On the SEM images of sample I, it is possible to observe a change in the surface morphology of the material, initiated by the action of a plasma DCSBD discharge on the sample surface. New, misaligned formations of various shapes and sizes have formed on the sample surface. It can be stated (Figure 12 at 2000× magnification) that the surface is dotted with globular formations. They act optically in such a way, that they tend to coalesce in certain places. When observing the surface at 50,000× magnification, formations with a size of 100–500 nm are visible, which are surrounded by smaller formations with a size in the order of a few nanometres. This finding in the surface of the investigated material may explain the formation of white matte stripes on the sample surface, characteristic of each series of samples (I, II, III, IV), according to the type of exposure to the electrode winding in the ceramic dielectric. The emergence of these new entities made the material more matte. It can be assumed that the light passing through the material is scattered at the interface of these newly formed formations, whereby the material appears dull/matte in places and localities close to the occurrence of these white stripes.

On the SEM images of sample II, it is also possible to observe the occurrence of globular formations, on the other hand, it should be added that they have a more continuous character (globular formations merge into larger units than in sample I, nor are they encased in smaller formations than in sample I). In some cases, it seems as if up to three formations have been joined into one, which may be due to a longer exposure time to the plasma discharge, or to pulling a portion of the PVC sample closer to the electrode surface. Furthermore, it is possible to observe (in Figure 13 at 2000× and 6000× magnification) the areas of the surface where these formations occur in smaller numbers than usual. In this case, these are probably areas that have not been so exposed to the plasma discharge or have been exposed to the discontinuous action of DCSBD plasma. This could be caused by attracting or pulling a thin polymer film from the surface of the ceramic dielectric.

The SEM images of sample III (Figure 14) at magnifications of 2000×, 6000×, 15,000× and 50,000× are similar in their morphology to series I and II. It is possible to observe the occurrence of an area with a larger but also a lower occurrence of globular formations as well as the joining of several formations into one, as in sample I. In addition, the coating/surrounding these formations can be observed in this sample (image at 50,000× magnification) to the same extent as in sample I.

On the SEM images of sample IV (Figure 15) (at 2000×, 6000×, 15,000× and 50,000× magnification) new formations were observed merging into asymmetrical formations (images at 15,000× and 50,000× magnification), which in some cases reach the size of several hundred nanometres. This may be due to the characteristic type of modification, i.e., may result from the relationship between the location of the polymer film and the electrode winding. The result of the action on each surface was the creation of a pattern resembling the letter X. This means that areas in this shape have been exposed to the same type of plasma and electric discharges from both sides of the surface. This resulted in a certain synergy of heat and electric discharge generated directly from the electrode. In contrast to samples I and III, globular formations and smaller formations surrounding them occur to a lesser extent in this case.

Based on the SEM images observations and evaluations at magnifications of 2000×, 6000×, 15,000× and 50,000× of PVC sample surfaces modified using DCSBD plasma discharge, the following conclusions can be stated: In general, SEM analysis has shown that exposing the surface of a PVC film to a plasma discharge creates several formations on the material surface in various sizes and shapes (filamentary and diffuse). The generation of these formations takes place by initiating isolated segments, which are merged into groups, from which they merge into oval formations, and by the same mechanism into large aggregates. The growth mechanism is not applied over the entire surface, but selectively, as if randomly in selected places. This may be related to the filamentary nature (Figure 6) of the plasma discharge, influenced by the memory effect [65,66] and the type of dielectric used. It is reasonable to assume that in places where formations are formed, the intensity of micro-discharges will be high, and their rate will be the engine of the growth of formations. The fact that the surface morphology changes significantly, and the surface area increases, will also lead to a change in surface roughness [67,68,69].

The result of the surface modification of polymers using DCSBD plasma is thus affected by:Experimental setup:○position and location of the material according to the electrode winding in the ceramic dielectric (Figure 1 and Figure 3); the degree of the ceramic dielectric coverage by the modified material,○plasma exposure time—not investigated in this study, ○plasma reactor power—not investigated in this study,○the distance of the material from the plasma-generating ceramic dielectric—not investigated in this study.The process of plasma-chemical treatment, the result of which depends on:○The ratio of diffuse and filamentary plasma that interacts with the material:PVC film exposed to DCSBD plasma loses insulating properties; thanks to the transparency of PVC, it was possible to observe a homogeneous plasma layer interacting with the PVC surface facing the ceramic dielectric during the modification/activation process, as well as individual micro-discharges—as insulated electric arcs breakthroughs turned away from the ceramic dielectric.The effect of DCSBD plasma was observed by changing the contact angles (Table 1, Figure 4), changes in roughness and surface area increase using the AFM method (Figure 8, Figure 9 and Figure 10, Table 2), XPS analysis (Table 3) and SEM microscopy (Figure 11, Figure 12, Figure 13, Figure 14 and Figure 15).○The presence of ozone (Figure 5) generated during plasma modification and its concentration (this depends on all known variables listed in the experimental settings above + atmospheric conditions—atmospheric pressure, relative humidity in the laboratory/industrial application)—were not examined in this study.According to the available literature, ozone interacts with polymer double bonds. This reaction usually results in the breakdown of the polymer chain into fragments, which reduces the molecular weight of the individual chains. The material thus loses overall strength and other mechanical properties [39]. This fact is described below (Static tensile test—Table 4).○The intensity of UV radiation (especially UV-B) generated by plasma, affects the material during exposure.According to the literature, due to the presence of abnormalities in the polymer matrix caused by the presence of C=O and O-O, PVC shows the ability of photo-oxidative degradation. Evidence of degradation by photooxidation is cracking, embrittlement, yellowing and opacity of the polymer.

### 3.2. Study of Thermal Properties

#### 3.2.1. Determination of Glass Transition Temperature by DMA Analysis

Dynamic mechanical analysis, as the most sensitive thermal analysis capable of determining glass transition temperatures (E′: Figure 16; E″: Figure 17, Tan δ: Figure 18) of the material, made it possible to notice these temperatures in thin polymer films and to determine their change caused by the DCSBD plasma modification type. To determine the glass transition temperature from the curve of the Storage modulus (E′), it was necessary to determine the temperature referred to as “OnSet”. In plasma unmodified PVC polymer films, the glass transition temperatures increase in order from the E′ (54.5 °C), through the Loss modulus (E″) (64.4 °C) to the temperatures for Tan δ (72.5 °C). Such a sequence of increasing the glass transition temperatures across the modules of dynamic mechanical analysis is thus, in accordance with the literature [70,71,72,73,74] and the same order was recorded even after DCSBD plasma modification, specifically 56.1 °C, 63.9 °C and 72.9 °C, respectively (Table 4).

The average change in the glass transition temperature for the plasma modified PVC polymer film was an increase of 1.6 °C for the E′, a decrease of 0.5 °C for the E′′ and an increase of 0.5 °C for the Tan δ glass transition temperature (Figure 19). Thus, including the results of the standard deviations of the glass transition temperatures, it can be stated that the DCSBD plasma does not demonstrably influence the glass transition temperatures. Therefore, the statistical one-way ANOVA test was applied in the evaluation. The results, when the most significant differences were recorded between the individual measurements, are in Table 5.

The calculated value of F (*p* < 0.10) within the groups of measurements indicates that when comparing the glass transition temperatures of the Elastic modulus, the variance is:8.5 times larger when comparing samples II vs. III, both of which were plasma modified (transverse and oblique),3.9 times larger when comparing PVC x III samples3.5 times larger when comparing PVC x I samples. Both series compare the unmodified PVC polymer film with the selected type of plasma modification.

The calculated value of F (*p* < 0.10) within the groups of measurements indicates that when comparing the glass transition temperatures of the loss modulus, the variance is:5.8 times larger when comparing the loss modulus of plasma modified I x III samples.

The calculated value of F (*p* < 0.10) within the measurement groups indicates that when comparing the temperatures of the Tan delta glass transition, the variance is:3.6 times larger when comparing plasma modified II x III samples.

The application of a higher frequency in the dynamic mechanical analysis will cause an increase in the glass transition temperature of the PVC polymer film. At a lower frequency, the increase in this temperature will be less significant than at a higher frequency. The glass transition temperature of the PVC polymer film was measured as the temperature Tan δ_max_ at a frequency of 1 Hz = 66.10 °C and at a frequency of 10 Hz = 72.51 °C. In principle, the lower Tan δ values of the PVC polymer film after plasma modification indicate a higher proportion of the viscous component in the material. Due to the effect of external forces, the polymeric PVC films modified with DCSBD plasma are more likely to undergo permanent deformation [69]. 

#### 3.2.2. Determination of Glass Transition Temperature Using DSC Analysis

The glass transition temperatures of PVC polymer films before and after exposure to plasma DCSBD were determined by the DSC method. The total average change in the glass transition temperature of the polymer film after plasma exposure to DCSBD thus represents an increase of 0.49 °C. The average glass transition temperatures (Figure 20), by type of exposure to plasma, were recorded as follows: PVC (69.67 °C ± 0.48), I (69.58 °C ± 0.42), II (69.59 °C ± 2.27), III (70.68 °C ± 1.0) and IV (70.80 °C ± 0.35). Using the DCS method, the glass transition temperature was recorded as the “midpoint” temperature for the standard PVC sample = 69.67 °C [75,76]. The most significant increase in the glass transition temperature was found in samples III and IV (1.02 °C and 1.13 °C, respectively). 

The effect of DCSBD plasma (average value of all exposure types I, II, III, IV) on the change in the glass transition temperature of the PVC polymer film was first recorded, both in the same way for the DMA method and the DSC method: ○using dynamic mechanical analysis, this change represented, on average, an increase in the glass transition temperature by 0.39 °C,○using the DSC analysis, this change represented, on average, an increase of 0.49 °C.

### 3.3. Study of Mechanical Properties

By comparing the average values of tensile strength of plasma-treated PVC polymer films (exposures I, II, III, IV) with a reference sample (PVC unmodified), the following conclusions can be stated. The reference PVC sample achieves a tensile strength of 54.9 MPa. The tensile strength of PVC polymer films exposed to plasma by the type of exposure decreased on average by 1.1 MPa (difference between average value by exposure type and plasma non-treated PVC) to an average value of 53.8 MPa. Thus, the surface treatment of PVC film with plasma did not cause a significant change in the strength characteristics of most of the investigated materials. From the above, it can be assumed that the DCSBD plasma modified only the surfaces of the investigated materials, except for sample IV. In this case, the largest decrease to 51.7 MPa was measured for tensile strength. This may mean that surface modification by plasma in this way of exposure may cause the degradation of the polymer chains in the structure of the PVC polymer film. This change was initiated by UV radiation, which was absorbed by the PVC bonds [77]. The effect of radiation is usually amplified by the simultaneous action of oxygen, which leads to the increased absorption of UV radiation [78].

The plasma surface treatment of the PVC film did not cause a significant change in the Elongation of the material [79,80]. The reference sample together with samples I, III and IV showed a value of Elongation almost identical = ~3.5%. However, the Elongation value of the type II plasma-treated sample increased minimally (0.17%) and increased to 3.67% Table 4.

## 4. Conclusions

Given study provides new valuable findings quantified at the level of surface changes of PVC polymer films: DCSBD plasma exposure indirectly modified surfaces that were not primarily exposed to plasma. These findings were confirmed by a decrease in contact angles, changes in roughness and increases in surface area (AFM analysis), in addition to observations of new globular formations on surfaces by SEM analysis and monitoring of the changes in the elemental composition using XPS analysis, such as the decay of C–Cl bonds. The data in the present study were obtained from a large database based on many polymeric PVC films (modified by DCSBD plasma according to the scheme indirectly; directly-MONO; DUAL in different types of exposure: I, II, III and IV = exposure considering the position of the electrode winding in the ceramic dielectric generating the homogeneous plasma layer). The actual exposure type of the PVC polymer film on a ceramic dielectric may create a presumption of internal restructuring. The static tensile test showed a minimal decrease in the tensile strength of the DCSBD plasma modified PVC polymer film. The presented work also provides another new finding: the surface modification (especially its realization, exposition I, II, III and IV) may affect the glass transition temperatures of the material very slightly. It was thus possible to notice a significant increase in the glass transition temperature of up to 3.6 °C for the Storage modulus and the most significant decrease in the glass transition temperature of 1.8 °C for the Loss modulus. The changing viscoelastic properties of the material can be caused by the context of the surface modification, respectively, due to the side effects in the DCSBD plasma generation, applied primarily for the purpose of surface modification. In addition, the change in the glass transition temperature of the DCSBD plasma modified PVC polymer film was also measured by the DCS thermal method, where the change represented a 0.49 °C increase in the glass transition temperature. All recorded changes were realized despite the fact that there was no gap between the plasma-generating ceramic dielectric and the investigated polymer film.

## Figures and Tables

**Figure 1 materials-15-04658-f001:**
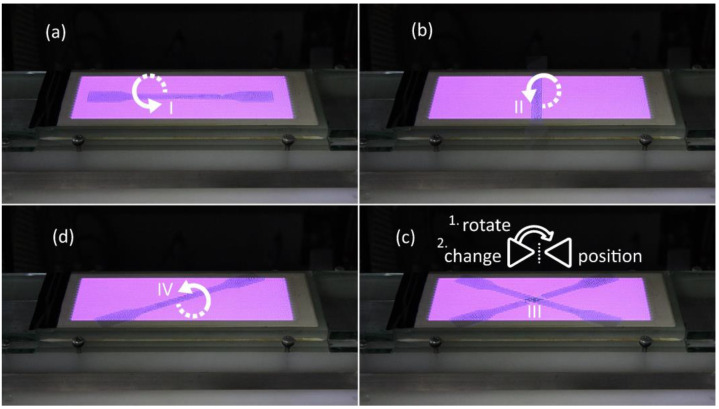
Side view of the plasma treatment films of PVC type (**a**) I, (**b**) II, (**c**) III, (**d**) IV.

**Figure 2 materials-15-04658-f002:**
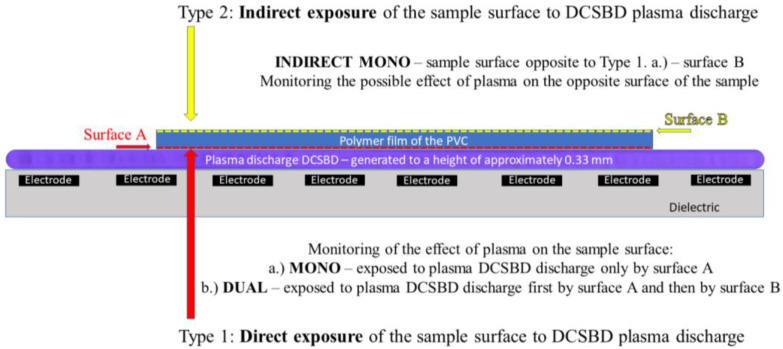
Type of PVC film exposure to the plasma DCSBD discharge according to the examined side of the sample (important in proving change of contact angles, AFM analysis and XPS analysis) and overall visualization of the electrode winding inside the ceramic dielectric generating the DCSBD plasma layer. Front view.

**Figure 3 materials-15-04658-f003:**
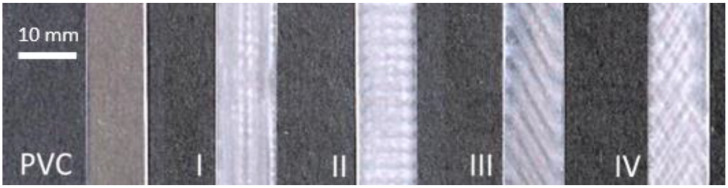
Investigated PVC polymer films: Without DCSBD plasma modification (PVC); Exposed to DCSBD plasma modification (I, II, III and IV). Characteristic bands are due to electrode wending in ceramic dielectric. Top view.

**Figure 4 materials-15-04658-f004:**
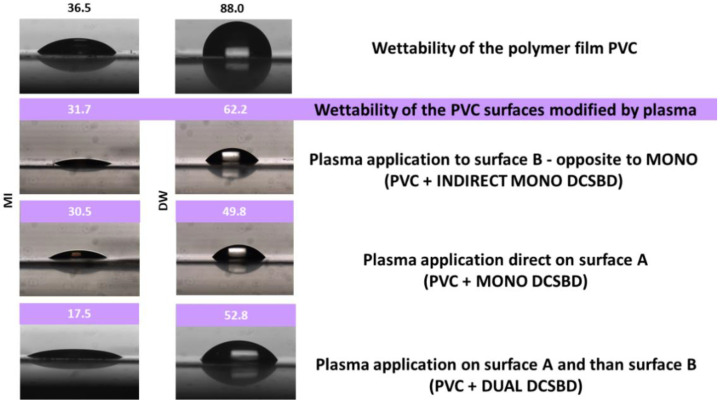
Wettability of PVC surfaces (unmodified/modified by plasma: Type 1/Type 2).

**Figure 5 materials-15-04658-f005:**
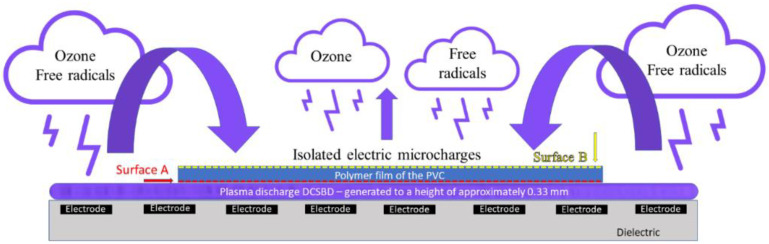
Fallout of active particles that are generated and released from the free surface of the ceramic dielectric during the generation of a DCSBD plasma discharge. This is followed by an interaction with the averted surface of the modified material, causing an indirect modification of surface B. Probably, magnetic field alongside electrodes/ceramic dielectric supports this effect. Front view.

**Figure 6 materials-15-04658-f006:**
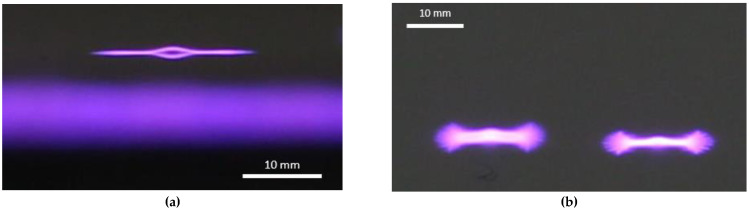
Insulated electric micro discharge generated on the ceramic dielectric of a KPR plasma reactor 20 (front view) and the shape of the electric micro discharges. (view at 15° and 45° angle), (**a**) front view at 15° angle, (**b**) front view at 45° angle.

**Figure 7 materials-15-04658-f007:**
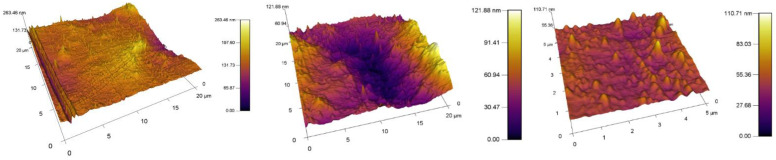
Different topography of unmodified surfaces of PVC.

**Figure 8 materials-15-04658-f008:**
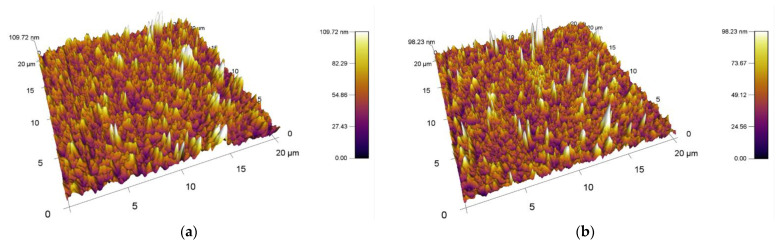
AFM images of plasma modified (modification type 1: DUAL) surfaces of PVC. From the left exposure types, (**a**) I, (**b**) II, (**c**) III, (**d**) IV.

**Figure 9 materials-15-04658-f009:**
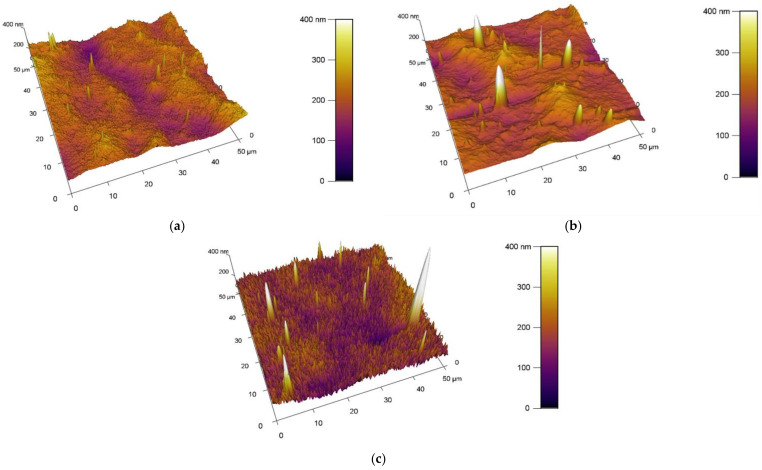
AFM images of plasma modified (modification type 1: MONO) surfaces of PVC. Exposure types I (**a**), II (**b**) and III (**c**).

**Figure 10 materials-15-04658-f010:**
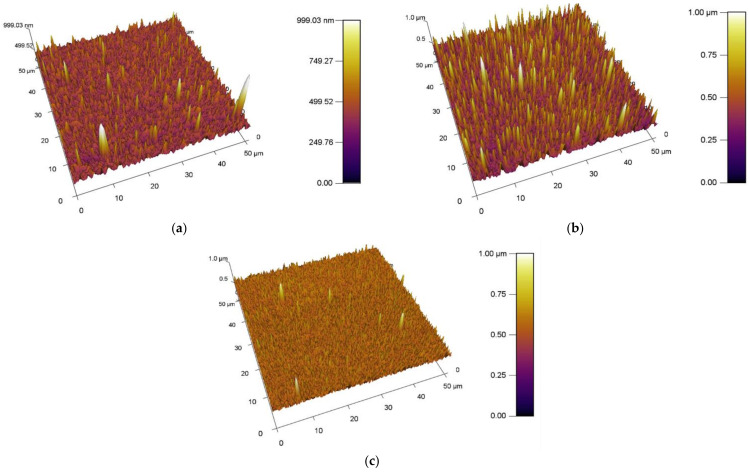
AFM images of plasma modified (modification type 2: INDIRECT MONO) surfaces of PVC. Exposure types I (**a**), II (**b**) and III (**c**).

**Figure 11 materials-15-04658-f011:**
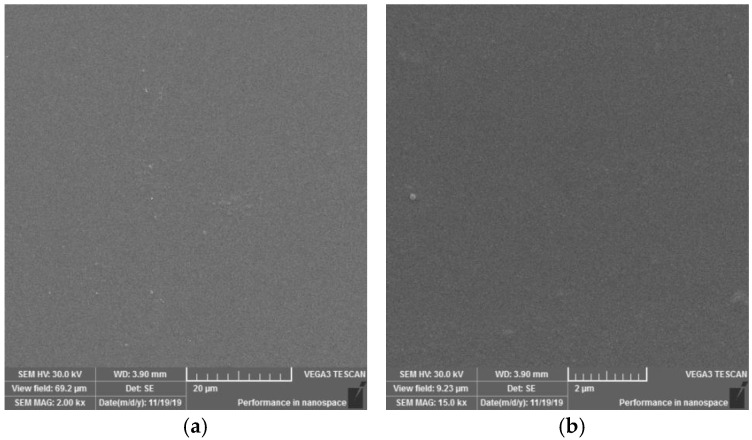
SEM morphology of the surface of unmodified PVC. Observed magnifications: 2000× (**a**) and 15,000× (**b**).

**Figure 12 materials-15-04658-f012:**
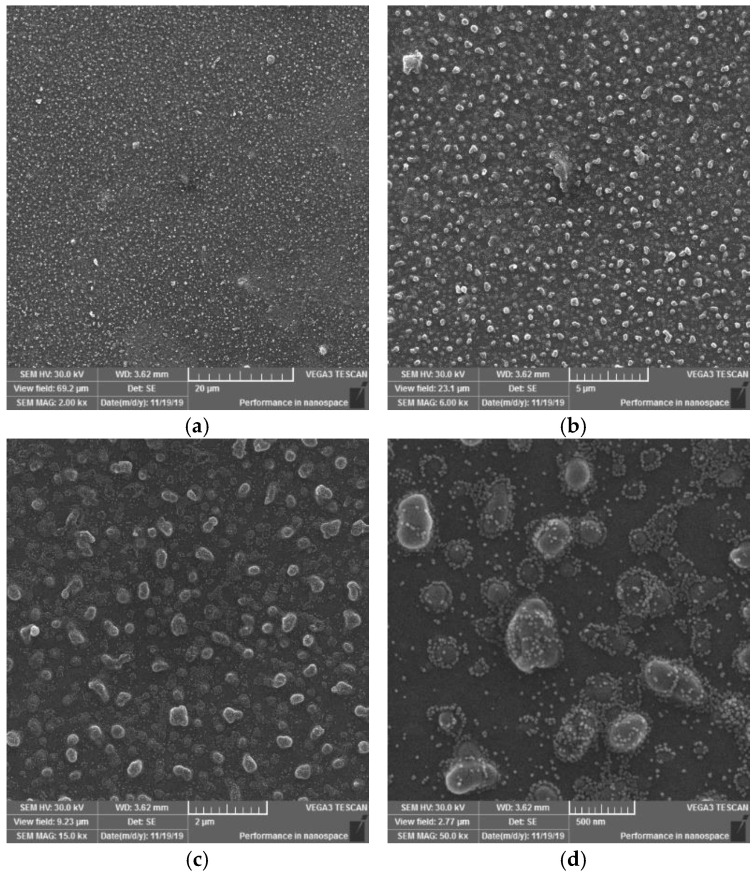
SEM morphology of the surface of plasma modified PVC exposure type I. Observed magnifications: 2000× (**a**), 6000× (**b**), 15,000× (**c**), 50,000× (**d**).

**Figure 13 materials-15-04658-f013:**
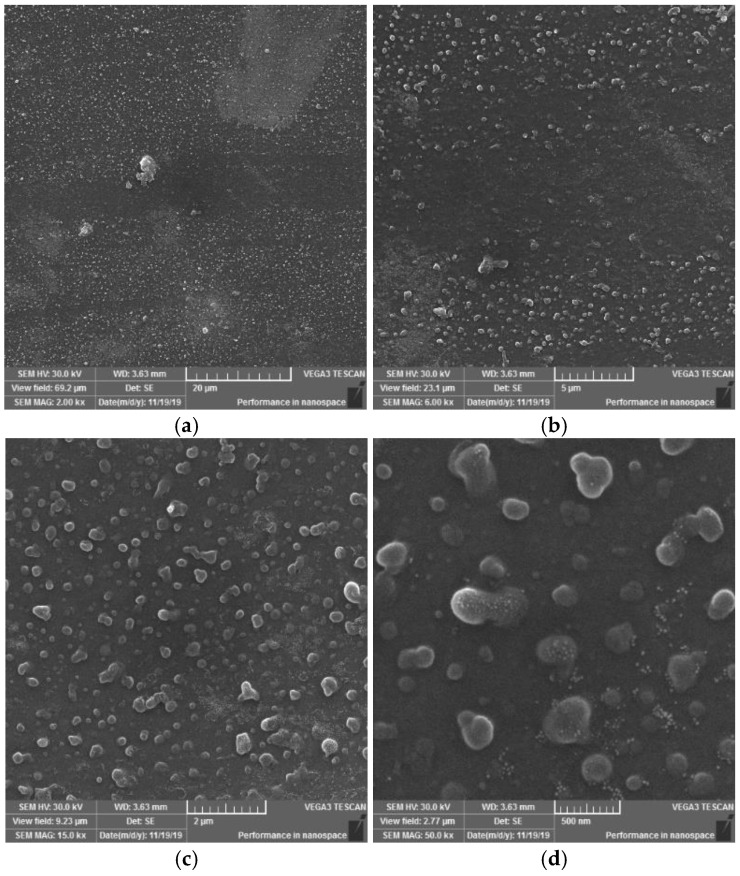
SEM morphology of the surface of plasma modified PVC exposure type II. Observed magnifications: 2000× (**a**), 6000× (**b**), 15,000× (**c**), 50,000× (**d**).

**Figure 14 materials-15-04658-f014:**
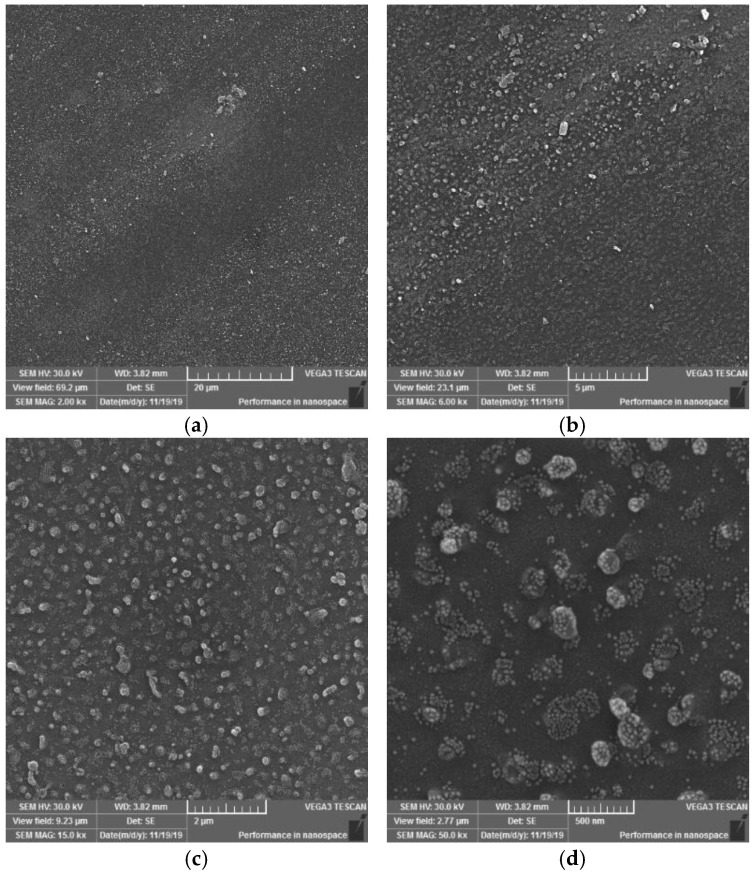
SEM morphology of the surface of plasma modified PVC exposure type III. Observed magnifications: 2000× (**a**), 6000× (**b**), 15,000× (**c**), 50,000× (**d**).

**Figure 15 materials-15-04658-f015:**
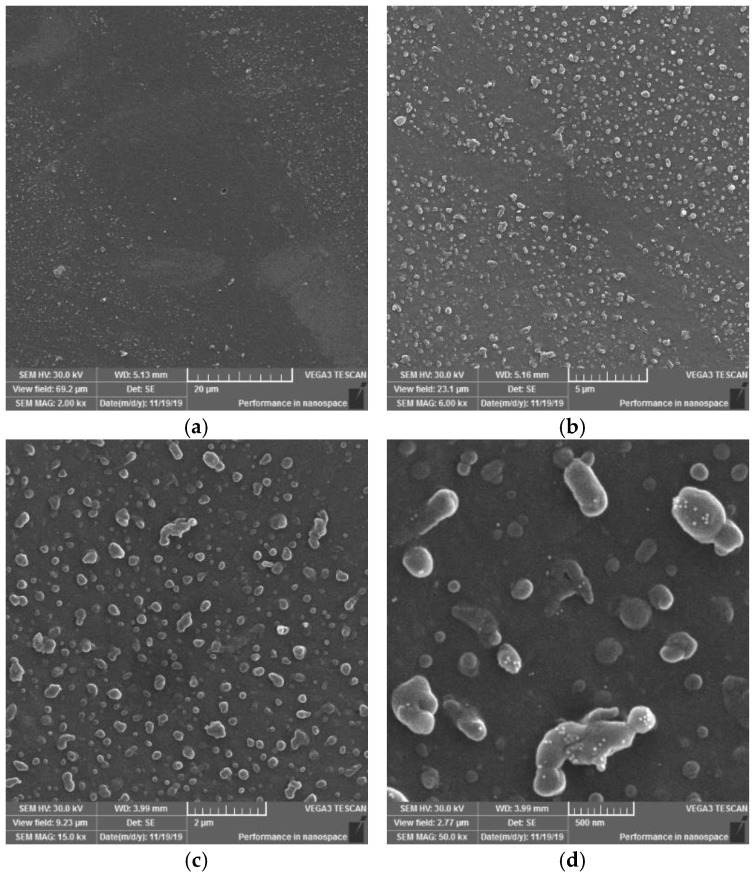
SEM morphology of the surface of plasma modified PVC exposure type IV. Observed magnifications: 2000× (**a**)**,** 6000× (**b**), 15,000× (**c**), 50,000× (**d**).

**Figure 16 materials-15-04658-f016:**
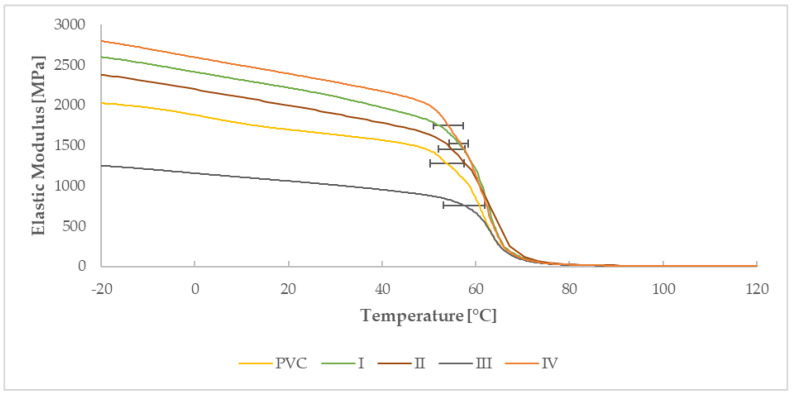
Determined curves of the elastic modulus of the PVC, I, II, III, IV sample.

**Figure 17 materials-15-04658-f017:**
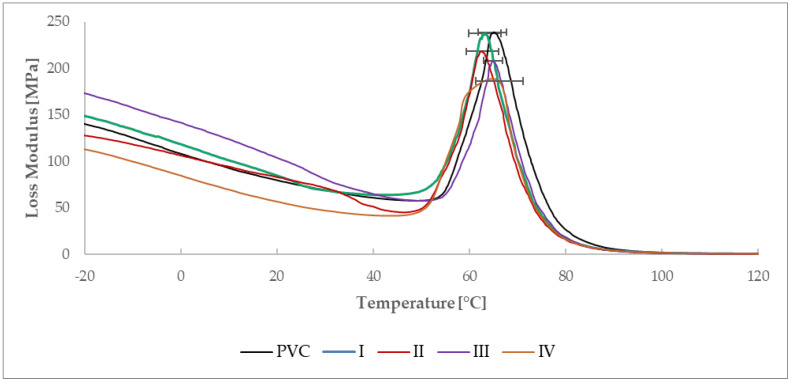
Determined curves of the Loss modulus of the sample PVC, I, II, III, IV.

**Figure 18 materials-15-04658-f018:**
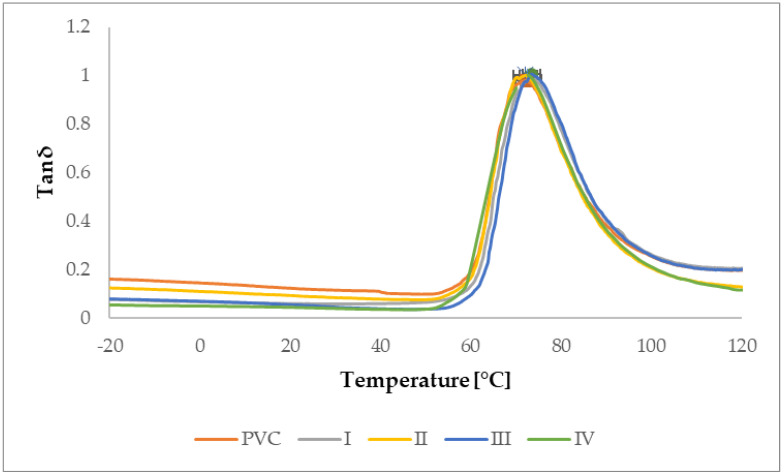
Determined Tan Delta curves of PVC samples, I, II, III, IV.

**Figure 19 materials-15-04658-f019:**
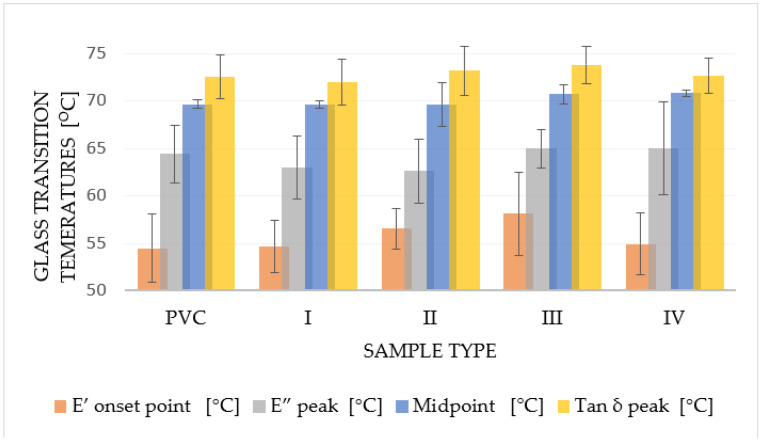
Graphical comparison of changes in glass transition temperature of plasma unmodified PVC and series of plasma modified polymer films PVC (I–IV) for individual modules of dynamic-mechanical analysis E′ and E′′, Tan δ and glass transition temperature obtained from DCS analysis (Midpoint).

**Figure 20 materials-15-04658-f020:**
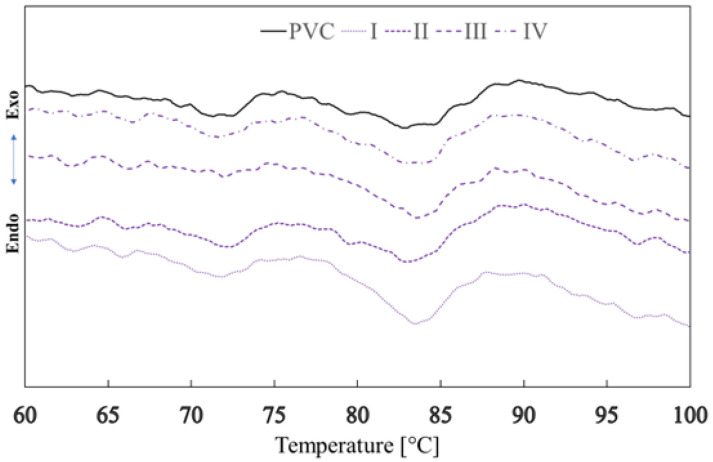
DSC curves for PVC and plasma treated PVC samples I, II, III a IV.

**Table 1 materials-15-04658-t001:** Contact angle values of samples using the DUAL, MONO, and INDIRECT MONO modifications.

Sample Type Sample Designation			Test Liquid					
			MI	DW	MI	DW	MI	DW
			Modification Type					
			Type 1: Direct Exposure				Type 2: Indirect Exposure	
			MONO		DUAL		INDIRECT MONO	
			Contact Angle (°)					
Unmodified		PVC	36.5 ± 2.7	88.0 ± 1.7	36.5 ± 2.7	88.0 ± 1.7	36.5 ± 2.7	88.0 ± 1.7
DCSBD plasma modified	By exposure type	I	30.2 ± 4.9	49.4 ± 4.7	20.3 ± 2.6	51.8 ± 1.9	30.7 ± 7.5	64.9 ± 8.1
		II	28.9 ± 4.2	50.2 ± 3.0	20.1 ± 1.8	50.0 ± 2.0	33.2 ± 4.9	62.9 ± 4.5
		III	32.4 ± 3.3	49.9 ± 5.4	16.0 ± 1.1	55.3 ± 0.7	31.2 ± 4.1	58.9 ± 6.3
		IV	*	*	13.5 ± 2.0	53.9 ± 1.6	*	*
	Average **		30.5	49.8	17.5	52.8	31.7	62.2
	Difference ***		6.0	38.2	19.1	35.3	4.8	25.8
	Difference **** [%]		Reference modification		−54.7%	−57.8%	3.8%	24.9%

* Unable to prepare this type of sample Best modification homogenity; ** Average value by exposure type The most significant decrease in the value of the contact angle; *** Difference between PVC and average value by exposure type; **** Difference between modification type-Type 1 MONO and Type DUAL, respectively, INDIRECT MONO.

**Table 2 materials-15-04658-t002:** Comparison of surface area increase and change of roughness on PVC polymer film surfaces after different types of modification using DCSBD plasma discharge.

			Type 1: Direct Exposure				Type 2: Indirect Exposure	
Sample Type Sample Designation			MONO		DUAL		INDIRECT MONO	
			Increase in Area [%]	Roughness *S_a_* [nm]	Increase in Area [%]	Roughness *S_a_* [nm]	Increase in Area [%]	Roughness *S_a_* [nm]
Unmodified		PVC	0.04 ± 0.00	5.98 ± 0.15	0.04 ± 0.00	5.98 ± 0.15	0.04 ± 0.00	5.98 ± 0.15
DCSBD plasma modified	by exposure type	I	3.91 ± 1.38	37.02 ± 3.18	0.81 ± 0.03	10.28 ± 0.29	0.49 ± 0.56	24.93 ± 2.89
		II	4.66 ± 1.29	38.69 ± 3.50	1.12 ± 0.02	15.11 ± 0.47	0.25 ± 0.20	22.014 ± 4.28
		III	4.05 ± 1.03	36.36 ± 3.35	9.37 ± 0.39	37.38 ± 2.72	0.06 ± 0.01	4.28 ± 0.57
		IV	*		12.05 ± 1.17	52.98 ± 4.66	*	
	Average **		4.21	37.36	5.84	28.94	0.27	17.7
	Difference ***		4.17	31.38	5.80	22.96	0.23	11.9

* Unable to prepare this type of sample; ** An average value by exposure type; *** Difference between PVC and average value by exposure type.

**Table 3 materials-15-04658-t003:** Representation of elements on selected samples/types of modifications.

Sample Type Sample Designation			Type 1: Direct Exposure (DUAL)					
			Element (%)					
			C 1s		O 1s		Cl 2p	
			Observed Area					
			Clear	Matte	Clear	Matte	Clear	Matte
Unmodified		PVC	77.6	*	12.6	*	7.1	*
DCSBD plasma modified	By exposure type	I	60.8	60	17.2	20.4	19.2	17.2
		II	62.1	60.4	18.1	19.2	18.2	17.1
		III	61.2	60.2	17.8	19.1	17.9	17.4
		IV	60.5	59.9	19.1	19.4	18.5	17.6
	Average **		61.2	60.1	18.1	19.5	18.5	17.3
	Difference ***		−16.5	-	5.5	-	11.4	-
			Type 2: Indirect exposure					
	By exposure type	I	61.8	61.3	18.9	18.6	17.3	17.2
	Difference ****		0.6	1.2	0.9	−0.9	−1.2	−0.1

* The matte area is formed only after plasma application; ** An average value by exposure type; *** Difference between PVC and average value by exposure type; **** Difference between average value of DUAL and INDIRECT MONO.

**Table 4 materials-15-04658-t004:** Tensile test results: dynamic-mechanical analysis (glass transition temperatures) with Tan δ values and static tensile test (Tensile strength and Elongation).

Dynamic-Mechanical Analysis (DUAL)							Static Tensile Test (DUAL)	
Sample Type Sample Designation			Glass Transition Temperature-T_g_ [°C]			Tan δ_value_	Tensile Strength [MPa]	Elongation [%]
			E′ Onset Point	E″ Peak	Tan δ_max_			
Unmodified		PVC	54.5 ± 3.6	64.4 ± 3.0	72.5 ± 2.3	0.9880 ± 0.019	54.9 ± 2.1	3.5 ± 0.2
DCSBD plasma modified	By exposure type	I	54.7 ± 2.8	63.0 ± 3.3	72.0 ± 2.4	0.9790 ± 0.015	54.9 ± 1.6	3.5 ± 0.1
		II	56.5 ± 2.1	62.6 ± 3.4	73.2 ± 2.6	0.9832 ± 0.008	54.5 ± 1.7	3.7 ± 0.2
		III	58.1 ± 4.4	65.0 ± 2.0	73.8 ± 2.0	0.9782 ± 0.033	54.0 ± 2.2	3.5 ± 0.1
		IV	54.9 ± 3.2	65.0 ± 4.9	72.7 ± 1.9	0.9915 ± 0.019	51.7 ± 1.6	3.5 ± 0.1
	Average *		56.1	63.9	72.9	0.9829	53.8	3.6
	Difference **		1.6	−0.5	0.4	−0.0051	−1.1	0.05

* An average value by exposure type; ** Difference between PVC and average value by exposure type.

**Table 5 materials-15-04658-t005:** Results of One Way ANOVA test of glass transition temperatures from DMA analysis.

**One Way ANOVA Test (*p* < 0.10)** **for Glass Transition Temperatures (PVC) and after Plasma Modification (I, II, III, IV)** **Mutual Comparison (x)**						
**Sample**	**E′**		**E″**		**Tan δ_max_**	
	**F**	** *p* **	**F**	** *p* **	**F**	** *p* **
PVC x I	3.49	0.10	0.98	0.35	0.77	0.40
PVC x II	0.02	0.90	1.71	0.23	0.86	0.38
PVC x III	3.90	0.08	0.20	0.67	0.97	0.97
PVC x IV	0.01	0.92	0.05	0.84	0.27	0.62
**One Way ANOVA Test (*p* < 0.10)** **for Glass Transition Temperatures of Plasma Modified Samples I, II, III, IV** **Mutual Comparison (x)**						
I x II	2.10	0.18	0.06	0.82	0.77	0.40
I x III	0.80	0.40	5.80	0.04	0.14	0.72
I x IV	1.37	0.27	1.07	0.33	0.12	0.74
II x III	8.48	0.02	3.07	0.12	3.58	0.09
II x IV	0.05	0.83	0.90	0.37	0.50	0.50
III x IV	2.08	0.18	0.21	0.66	2.24	0.17

## Data Availability

Data are available upon request to the corresponding author.

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
