# Peer review of "DMA Analysis of Plasma Modified PVC Films and the Nature of Initiated Surface Changes"

_materials, 2022, doi:10.3390/ma15134658_

Round 1

Reviewer 1 Report

This manuscript investigated the effect of PVC polymer film modification by DCSBD plasma by characterizing  the Tg of the samples by DMA. In my opinion, the way the data were present did not support the finds/results reported, therefore I recommend to the Authors to revise the analysis the manuscript using One-Way ANOVA statistical test, to determine which types of treatment are significantly different from each other and from non-treated sample. Additional comments are:

·       1.  Improve the introduction part, describing the most relevant finds of at least two similar studies, not only mention them as reference.

·    2.   Materials and methods: “Five series of experimental samples of PVC polymer film were prepared for the experiment”. “Each series contained 10 samples die cut on a pneumatic die cutting machine into a standardized size and shape referred to as type 1B”. Selected test samples  for 4 series of PVC film were surface modified using DCSBD plasma.” The investigated  polymer is a rigid PVC foil standardly used in office applications. “  The previous sentences are not clear. Five or four series? What is type 1B? What is a rigid PVC foil standardly?

·    3. Materials and methods. Improve the description of DCSBD experimental method used.

·    4.    Materials and methods. A text comparing differences among angle contact test methods is not necessary in the materials and methods. The Authors just have to describe the test protocols used.

·    5.    Results and discussions. The first paragraph is a description of DCSBD experimental part, not the result.

    6. Results and discussions. Change size Figure 10 to be equal to the other SEM photos.

   7.  Results and discussions. Table 4 contains a mistake regarding the Tg nomenclature.

·  8.    Results and discussions. “The average change in the glass transition temperature for the plasma modified PVC polymer film was an increase of 1.6 °C for the E′, a decrease of 0.5 °C for the E′′ and an increase of 0.5 °C for the Tan δ glass transition temperature (Figure 15)”. This sentence is not true considering the error bar. Indeed, the Tg values did not changed after the modification by DCSBD, and the same conclusion can be drawn is DSC.

·    9.   Results and discussions .“Using the DCS method, the glass transition temperature was recorded as the "midpoint" temperature for the standard PVC sample = 69.67 °C [79, 80]. The most significant increase in the glass transition temperature was found in samples III and IV (1.02 °C  and 1.13 °C, respectively). This finding may thus be directly related to the change in the viscoelastic properties of the polymer films due to the exposure to the electrodes in the ceramic dielectric and the formation of new, matte white areas.”. The sentence is confused.

·     10.    Results and discussions. “The tensile strength of PVC polymer films exposed to plasma by the type of exposure decreased on average by 1.1 MPa to an average value of 53.8 MPa.” The code of type is missed. In fact, this reduction appears not significant considering the standard deviation.

Reviewer 2 Report

The authors investigate the effect of plasma treatment on water contact angle, glass transition temperature, surface roughness and chemical composition change of PVC films.

However, the authors do present substantial measurement, the paper fails to underline and express the significance of this work, and the novelty of the findings.

The following issues should be addressed:

Title: since the paper is about PVC the title should be more specific and include PVC. Since the glass transition does not change significantly by the plasma treatment the authors should reconsider the title.

Abstract: before using the abbreviation DCSBD please spell it out.

Introduction: The authors reference a number of articles; however the introduction is very short and does not present the state of the art of the particular research, the problem that they are trying to solve, and novelty of the works. Thus, the authors should rewrite this part completely. The major prior work should be presented in detail. A schematic of the plasma treatment setup should be included for better understanding.

line 66-74: Based on this description the experiments are not repeatable for other researchers. The source of the PVC and grade, MFI or other specific information should be given.

line 70: type 1B of what, a standard? please give specific dimensions, see the previous comment.

In the Materials and methods section manufacturer and country of origin should be included for all of the used equipment, like in line 144, alongside with the model.

line 75-80: the authors should mention what are the different parameters used for the four PVC samples.

line 99: do the authors refer to the tapping mode of AFM?

line 101-103: It is confusing what is the measured area, 20x20 um or 5x5 um?

Figure 1. The authors should specify what does PVC I, II, III and IV designations mean!

Line 79-93: the authors should state what values the cited literature found.

chapter 3.1: the authors should explain first what differences between each measurement are and then discuss the results.

Figure 6. Please state what is the difference between the AFM 3 graphs, different samples?

Figure 8 and 9: the area covered is 50x50 um. In the description 20x20 um was stated…

Table 4: it would be beneficial to show the actual E’, E’’ and tan delta graphs.

Figure 16: it would be beneficial to show the actual DSC thermograms.

Line 619-622: since the error bars are overlapping this statement is not supported.

Conclusion: the authors should underline the major findings of the work and express the utility of the findings.

Reviewer 3 Report

The paper on "Glass transition temperature changes of plasma modified polymer films" discusses about DCSBD plasma for the modification of viscoelastic property of the polymer films. 

1. Avoid abbreviations in the abstract or define them in the start. 

2. Explain the mechanism of the changes happening in the physical properties of the films, what chemical structure evaluation using chemical composition analysis.

3. English needs some degree of improvement, as the structure is not harmonious. 

4. AFM analysis is not conclusive, its better to measure roughness (RHM).

Round 2

Reviewer 1 Report

 The revison should to be improved. My comments are:

1)    Results and discussions. Table 4 contains a mistake regarding the Tg nomenclature.

2)    Results and discussions. “The average change in the glass transition temperature for the plasma modified PVC polymer film was an increase of 1.6 °C for the E′, a decrease of 0.5 °C for the E′′ and an increase of 0.5 °C for the Tan δ glass transition temperature (Figure 15)”.

Again, this sentence is not true Indeed, the ANOVA analysis included by the Authors supports the statement that Tg does not change since the P-values are higher than 0,1, which means that there is no statistical difference between the groups.  The Authors have to review this part.

Reviewer 3 Report

Accept as it is. 

Author Response

Thank you for your valuable comments that have helped improve the quality of our work and your positive feedback on our work. 

Round 3

Reviewer 1 Report

Dear Editor,

The changes were properly made, so I recommend accepting the manuscript.